

# Rhizobacterial communities of five co-occurring desert halophytes

Yan Li[1,2,3], Yan Kong[4,5], Dexiong Teng[2], Xueni Zhang[1,2], Xuemin He[1,2], Yang Zhang[6] and Guanghui Lv[1,2]

[1] Key Laboratory of Oasis Ecology of Education Ministry, Xinjiang University, Urumqi, Xinjiang, China
[2] Institute of Arid Ecology and Environment, Xinjiang University, Urumqi, Xinjiang, China
[3] Ecology Post-doctoral Research Station, Xinjiang University, Urumqi, Xinjiang, China
[4] School of Life Science and Biotechnology, Shanghai Jiaotong University, Shanghai, China
[5] SJTU-Yale Joint Center for Biostistics, Shanghai Jiaotong University, Shanghai, China
[6] College of Resource and Environment Sciences, Xinjiang University, Urumqi, Xinjiang, China

## ABSTRACT

**Background**. Recently, researches have begun to investigate the microbial communities associated with halophytes. Both rhizobacterial community composition and the environmental drivers of community assembly have been addressed. However, few studies have explored the structure of rhizobacterial communities associated with halophytic plants that are co-occurring in arid, salinized areas.

**Methods**. Five halophytes were selected for study: these co-occurred in saline soils in the Ebinur Lake Nature Reserve, located at the western margin of the Gurbantunggut Desert of Northwestern China. Halophyte-associated bacterial communities were sampled, and the bacterial 16S rDNA V3–V4 region amplified and sequenced using the Illumina Miseq platform. The bacterial community diversity and structure were compared between the rhizosphere and bulk soils, as well as among the rhizosphere samples. The effects of plant species identity and soil properties on the bacterial communities were also analyzed.

**Results**. Significant differences were observed between the rhizosphere and bulk soil bacterial communities. Diversity was higher in the rhizosphere than in the bulk soils. Abundant taxonomic groups (from phylum to genus) in the rhizosphere were much more diverse than in bulk soils. Proteobacteria, Firmicutes, Actinobacteria, Bacteroidetes and Planctomycetes were the most abundant phyla in the rhizosphere, while Proteobacteria and Firmicutes were common in bulk soils. Overall, the bacterial community composition were not significantly differentiated between the bulk soils of the five plants, but community diversity and structure differed significantly in the rhizosphere. The diversity of *Halostachys caspica*, *Halocnemum strobilaceum* and *Kalidium foliatum* associated bacterial communities was lower than that of *Limonium gmelinii* and *Lycium ruthenicum* communities. Furthermore, the composition of the bacterial communities of *Halostachys caspica* and *Halocnemum strobilaceum* was very different from those of *Limonium gmelinii* and *Lycium ruthenicum*. The diversity and community structure were influenced by soil EC, pH and nutrient content (TOC, SOM, TON and AP); of these, the effects of EC on bacterial community composition were less important than those of soil nutrients.

**Discussion**. Halophytic plant species played an important role in shaping associated rhizosphere bacterial communities. When salinity levels were constant, soil nutrients emerged as key factors structuring bacterial communities, while EC played only a

Corresponding author
Guanghui Lv, ler@xju.edu.cn

minor role. Pairwise differences among the rhizobacterial communities associated with different plant species were not significant, despite some evidence of differentiation. Further studies involving more halophyte species, and individuals per species, are necessary to elucidate plant species identity effects on the rhizosphere for co-occurring halophytes.

# INTRODUCTION

Salinization is a serious land degradation problem, as high salinity limits plant growth. Salts may accumulate in soils as a result of natural processes, such as mineral weathering, dust collection and precipitation, or artificial processes, such as irrigation (*Oosterbaan, 1988*); both may lead to saline soils that make it difficult for plants to absorb moisture from the soil. Halophytes are salt-tolerant plants that can grow in areas with salt (NaCl) concentrations higher than 400 mM (*Flowers, 2004*; *English & Colmer, 2011*). In saline soil environments, halophytes play an important role in carbon sequestration, nutrient mineralization, nutrient cycling and improvement of the micro-environment (*Cao et al., 2014*; *Chaudhary et al., 2015*), and may have great potential to preserve ecosystems.

Salinity tolerance in halophytes is not solely due to physiological mechanisms and their genetic regulation (*Vasquez et al., 2005*), but also to complex ecological processes within the plant rhizosphere and phyllosphere; microorganisms inhabiting the roots and leaves of halophytes may significantly contribute to their salinity tolerance (*Ruppel, Franken & Witzel, 2013*). Many microorganisms have plant growth-promoting activities and confer salt tolerance on halophytic plants (*Nabti et al., 2007*; *Sgroy et al., 2009*; *Jha, Gontia & Hartmann, 2012*; *Mapelli et al., 2013*). Recent studies of halophyte-associated microbial communities have addressed the ecological and environmental drivers underlying community assembly and recruitment (*Jha, Gontia & Hartmann, 2012*; *Borruso et al., 2014*; *Marasco et al., 2016*; *Chaudhary et al., 2017*; *Tian & Zhang, 2017*). These studies have revealed that bacterial communities in the halophyte rhizosphere are distinctly different from those of non-halophytic plants, containing a larger proportion of halophilic bacteria (*Al-Mailem et al., 2010*). Many halophilic bacteria have been identified or isolated from halophyte roots, soils and desert habitats, including species belonging to the following genera: *Alkalimonas*, *Bacillus*, *Brachybacterium*, *Brevibacterium*, *Cronobacter*, *Halobacillus*, *Halomonas*, *Marinococcus*, *Methylibium*, *Nesterenkonia*, *Oceanobacillus*, *Staphylococcus*, *Stenotrophomonas*, *Virgibacillus* and *Zhihengliuella* (*Sgroy et al., 2009*; *Siddikee et al., 2010*; *Tang et al., 2011*; *Shi et al., 2012*; *Zhou et al., 2012*; *Ramadoss et al., 2013*; *Borsodi et al., 2015*; *Zhao et al., 2016*).

The diversity and composition of the rhizosphere bacterial community depends not only on the plant species, but also on various soil parameters (*Tian & Gao, 2014*; *Rodriguez-Blanco, Sicardi & Frioni, 2015*; *Pii et al., 2016*; *Song et al., 2017*). Different plant species, or even genotypes within species, tend to assemble distinct rhizobacterial communities

(*Chaudhary et al., 2015*). For example, in the rhizosphere of *Aster tripolium*, Actinobacteria, Firmicutes and Proteobacteria are the most abundant bacterial phyla, and *Bacillus* the dominant genus (*Szymanska et al., 2016b*). In contrast, Acidimicrobiales, Myxococcales and Sphingomonadales are common in the rhizosphere of *Halimione portulacoides* and *Sarcocornia perennis* (*Oliveira et al., 2014*). The *Puccinellia limosa* rhizosphere is dominated by *Halomonas* and *Nesterenkonia* species (*Borsodi et al., 2015*). Similar rhizobacterial communities may be found in different environments when the same plant species is present (*Smalla et al., 2001*; *Berg & Smalla, 2009*). However, in certain environments, such as hypersaline soils, plant species identity plays only a minor role compared to soil salinity in shaping microbial community structure (*Borruso et al., 2014*); this result has been confirmed by studies of desert soils (*Li et al., 2013*). Overall, both plant species identity and soil type can be important, depending on abiotic and biotic conditions (*Berg & Smalla, 2009*).

While many studies have investigated the effects of plant species identity and various soil properties, as well as other factors (i.e., temperature, geographical distance), on structuring microbial communities in the rhizosphere, few studies have characterized the rhizobacterial communities associated with halophytic plants found in arid, saline environments. In a Mediterranean salt marsh in Southeastern Spain, a study of eight halophytes (*Asteriscus maritimus*, *Arthrocnenium macrostachyum*, *Frankenia corymbosa*, *Halimione portulacoides*, *Limonium cossonianum*, *Limonium caesium*, *Lygeum spartum* and *Suaeda vera Forsskal*) examined the soil microbiological and biochemical properties of the rhizosphere, revealing that soil microbial activity and microbial-related soil properties, such as aggregate stability, were determined by the plant species. However, the community composition of the microbes was not examined (*Caravaca et al., 2005*). Another study recently compared the microbial communities associated with three dominant halophytes (*Aeluropus*, *Salicornia* and *Suaeda*) in a coastal region of India (*Chaudhary et al., 2015*). To date, in halophytic species occurring in arid desert environments, little is known regarding rhizobacterial community assembly and the relative contributions of rhizosphere effects versus salinity to this process.

In this study, the bacterial communities associated with five halophytes (*Halocnemum strobilaceum*, *Halostachys caspica*, *Limonium gmelinii*, *Lycium ruthenicum* and *Kalidium foliatum*) growing in arid, saline environments were characterized. The diversity and structure of rhizosphere bacterial communities was investigated using an Illumina MiSeq sequencing approach. The study goals were to: (1) compare the bacterial communities of the rhizosphere with those found in bulk soil samples, in order to understand the effects of plant species identity on bacterial communities; (2) compare the rhizosphere community composition of the five halophytic species so as to look for similarities across species; and (3) evaluate the relative contributions of plant species identity and soil salinity in structuring rhizosphere bacterial communities in arid, saline habitats.

## MATERIALS & METHODS

### Study area and sample collection

Soil samples were collected from the Ebinur Lake Wetland, Xinjiang, China (44.595°N, 83.552°E) in July 2017, following previously established protocols (*Chaudhary et al., 2015*; *Edwards et al., 2015*). The Ebinur Lake Nature Reserve is located at the western margin of the Gurbantunggut Desert in Xinjiang, China. Conditions are windy in the Reserve, which has a typical dry, continental climate, with an annual average precipitation of 105 mm and evaporation of 1,315 mm. Soils in the Reserve are highly salinized and alkalized, with an average electrical conductivity (EC) of 5.41 mS/cm and pH of 8.77 in surface soils 0–10 cm deep; the mean soil water content is 7.19% (*Zhang, Yang & Lv, 2014*). As such, many halophytic species grow in this region. In this study, five co-occurring halophytic plants species, four shrubs (*Halocnemum strobilaceum*, *Halostachys caspica*, *Lycium ruthenicum* and *Kalidium foliatum*), and one perennial herb (*Limonium gmelinii*), were selected for study.

Thirty samples were collected in total: 15 from the rhizosphere and 15 paired bulk soil samples. Each rhizosphere sample came from a different plant; sampled individuals were distributed within a 1 km radius of the GPS coordinates provided above (Fig. S1). Three healthy individuals were randomly selected from each species to be sampled. Plant roots were dug up using a shovel, in order to collect all roots to a depth of approximately 35–45 cm. Excess soil was manually shaken from the roots, but any soil still attached after shaking (a layer ~1 mm thick) was retained for study. Each root sample was immediately placed into a sterile flask with 30 ml of sterile Phosphate Buffered Saline (PBS) solution (137 mmol/L NaCl, 2.7 mmol/L KCl, 8.5 mmol/LNa$_2$HPO$_4$, 1.5 mmol/L KH$_2$PO$_4$, pH 7.3). Bulk soil samples were collected from sites 30–40 cm away from the roots of a given plant; soils were collected to a depth of approximately 40 cm. About 100 g of fresh soil from each sample was stored in a sterile plastic bag; these were immediately transported back to the lab on ice. In the lab, the flasks containing root samples were stirred vigorously with sterile forceps to clean all the soil from the root surfaces. This soil was then poured into a 50 ml sterile Falcon tube ready for DNA extraction.

### Soil chemical analysis

Soil samples were dried in a hot air oven at 105 °C for 48 hours to determine the soil water content (SWC). After drying, samples were ground and sieved through 2 mm mesh. The electrical conductivity (EC) and pH were measured in a 1:2.5 (w:v) soil to water mix. The total organic carbon (TOC) and soil organic matter (SOM) were estimated using a spectrophotometer, after oxidizing soil samples with K$_2$Cr$_2$O$_4$ (*Yang, 1987*). The total nitrogen (TON) was determined using the Kjeldahl method (*Honda, 1962*). To measure available phosphorus (AP), samples were digested with HClO$_4$-H$_2$SO$_4$, and then the Mo-Sb colorimetric method was used for quantification (*Agrochemistry Committee of the Chinese Soil Society, 1983*).

## DNA extraction, amplification and sequencing

Rhizosphere samples were concentrated by pipetting 1 mL of the PBS/soil mix into a 2 mL sterile tube and centrifuging for 1 min at 10,000 g. The supernatant was discarded leaving only the soil fraction behind. About 0.2 g of soil (wet weight) from each bulk sample was transferred to a 2 mL sterile tube. Then, the total genomic DNA was extracted using an E.Z.N.A$^{TM}$ Mag-Bind Soil DNA Kit (OMEGA) following the manufacturer's instructions. DNA samples were inspected on a 1.0% agarose gel and quantified using a Nanodrop 2000 spectrophotometer (Nanodrop Technologies, Wilmington DE, USA). The bacterial 16S rDNA V3–V4 region was amplified and sequenced for analysis. PCR products were visualized using electrophoresis on 1.5% agarose gels and purified using VAHTS$^{TM}$ DNA Clean Beads (Vazyme, Nanjing, China). Finally, about 10 ng of DNA from each sample was sequenced on the Illumina MiSeq platform by Sangon Technology Co., Ltd. (Shanghai, China). The sequence data has been submitted to the NCBI Sequence Read Archive database under accession number SRP129060.

## Sequence preprocessing and OTU assignment

Quality control of raw sequencing data was conducted following Schmieder & Edwards (*Schmieder & Edwards, 2011*). Ambiguous bases with Phred quality score <20 at the end of a read and fragments containing Ns were trimmed. Reads with length <200 nucleotides (nt) were removed, and the remaining reads truncated to 400–450 nt sequences. Chimeric sequences were identified with UCHIME (*Edgar et al., 2011*) and discarded (109–1,397 chimeras per sample). The filtered sequences were then clustered into OTUs at a 97% similarity level. A representative sequence from each OTU was selected for both taxonomic annotation using the Ribosomal Database Project (RDP) classifier (*Wang et al., 2007*) and also to BLAST against the Silva and NCBI databases (*Quast et al., 2013*). OTUs with an RDP classification threshold below 0.8 or with identity and coverage lower than 90% were marked as unclassified. Singletons and sequences aligning to the mitochondria or chloroplast were removed (4–1,450 reads per sample). Finally, the number of sequences in each sample was normalized by random resampling to the smallest sample size ($n =$ 35,000) prior to calculation of species diversity indices (Shannon, Simpson, Chao1, ACE, and Good's coverage) using Mothur ver 1.30.1 (*Schloss et al., 2009*). OTU richness was calculated using the *vegan* package ver. 2.1-10 (*Dixon, 2003*) in R version 3.2. Rarefaction analysis was implemented in Mothur 1.30.1 and a rarefaction curve produced in R. Using R, a species accumulation curve was also constructed (with the *vegan* package) and diagrams depicting bacterial community structure (composition and relative abundance) at multiple taxonomic ranks (phylum, class, order, family and genus) were generated.

## Statistical analysis

One-way ANOVAs were used to test for differences in alpha diversity indices (OTU richness, Shannon diversity index and Good's coverage index) as well as soil physicochemical properties. A principal coordinates analysis (PCoA) was performed on weighted UniFrac distances (using the *vegan* package) to compare community composition among samples. A UPGMA tree depicting clustering relationships among samples was produced based on
**Table 1  Soil characteristics of bulk soil samples from five halophytes.**

|  | TOC (g/kg) | SOM (g/kg) | TON (g/kg) | AP (g/kg) | pH | EC (mS/cm) | SWC (%) |
|---|---|---|---|---|---|---|---|
| *Lycium ruthenicum* | 9.14 ± 3.43[a] | 15.75 ± 5.92[a] | 0.58 ± 0.24[a] | 0.89 ± 0.15[a] | 8.23 ± 0.37 | 5.56 ± 1.26[b] | 19.73 ± 2.18[a] |
| *Limonium gmelinii* | 10.78 ± 1.60[a] | 18.59 ± 2.77[a] | 0.60 ± 0.10[a] | 0.80 ± 0.08[a] | 8.33 ± 0.24 | 6.61 ± 0.91[ab] | 17.02 ± 3.51[a] |
| *Kalidium foliatum* | 11.27 ± 5.66[a] | 19.43 ± 9.76[a] | 0.64 ± 0.25[a] | 0.92 ± 0.08[a] | 8.02 ± 0.25 | 5.65 ± 0.53[b] | 16.45 ± 6.11[a] |
| *Halostachys caspica* | 5.53 ± 0.95[b] | 9.53 ± 1.63[b] | 0.34 ± 0.03[ab] | 0.82 ± 0.06[a] | 8.05 ± 0.16 | 6.78 ± 1.42[ab] | 10.42 ± 2.57[b] |
| *Halocnemum strobilaceum* | 3.15 ± 1.09[b] | 5.43 ± 1.88[b] | 0.21 ± 0.02[b] | 0.62 ± 0.06[b] | 8.05 ± 0.30 | 7.14 ± 1.46[a] | 17.71 ± 3.16[a] |
| mean | 8.05 ± 4.15 | 13.87 ± 7.15 | 0.48 ± 0.23 | 0.82 ± 0.14 | 8.14 ± 0.27 | 6.30 ± 1.21 | 16.40 ± 4.57 |

**Notes.**
[a] Values are given as means (± standard error) ($n = 3$).
[b] Different letters indicate significant differences among five halophytes at $P < 0.05$ level.

Bray-Curtis beta diversity distance metrics. ANOSIM (999 permutations) and Adonis statistics, available in the *vegan* package, were calculated to evaluate differences in rhizosphere community composition among the five study species. Venn diagrams illustrating similarities/differences in OTU composition between samples were produced with the *Venn-Diagram* package, ver. 1.6.16. Furthermore, STAMP 2.1.3 (*Parks et al., 2014*) and LEfSe 1.1.0 (*Segata et al., 2011*) were implemented to identify differentially abundant groups among samples. Pearson's correlation coefficients were calculated between the community diversity and richness indices and each soil property, and also among soil properties. Correspondence analysis (CA) was performed using the *vegan* package to investigate links between community structure, soil properties and plant species.

## RESULTS

### Soil properties

The average soil water content (SWC) of bulk soils was 16.40 ± 4.57%. The electrical conductance (EC) was 6.30 ± 1.21 mS/cm and pH 8.14 ± 0.27. The mean total organic carbon (TOC), soil organic matter (SOM), total nitrogen (TON) and available phosphorus (AP) were 8.05 ± 4.15 g/kg, 13.87 ± 7.15 g/kg, 0.48 ± 0.23 g/kg and 0.82 ± 0.14 g/kg, respectively. *Halocnemum strobilaceum* and *Halostachys caspica* soils had higher EC and lower TOC, SOM and TON compared to those of other species ($P < 0.05$). The AP content in *Halocnemum strobilaceum* soils was significantly lower than for other species ($P < 0.05$) (Table 1).

### Bacterial community diversity

In total, 1.83 Gb of raw sequence data was obtained from all samples; after quality controls, a total of 1.18 Gb of clean sequence reads were available for further analysis. Good's coverage for all samples was higher than 0.98 (Table S1). Rarefaction curves stabilized as the number of sequences increased (Fig. S2), suggesting that bacterial communities were reasonably well-characterized. Species accumulation curves nearly reached a plateau, where the number of OTUs did not increase with sample size, indicating that the sample size was sufficient for data analysis (Fig. S3). After removal of chimeras, plant sequences and singletons, a total of 1,315,341 reads were obtained from soil samples; these were grouped into 8,087 OTUs. OTU richness was higher in rhizosphere versus bulk soil

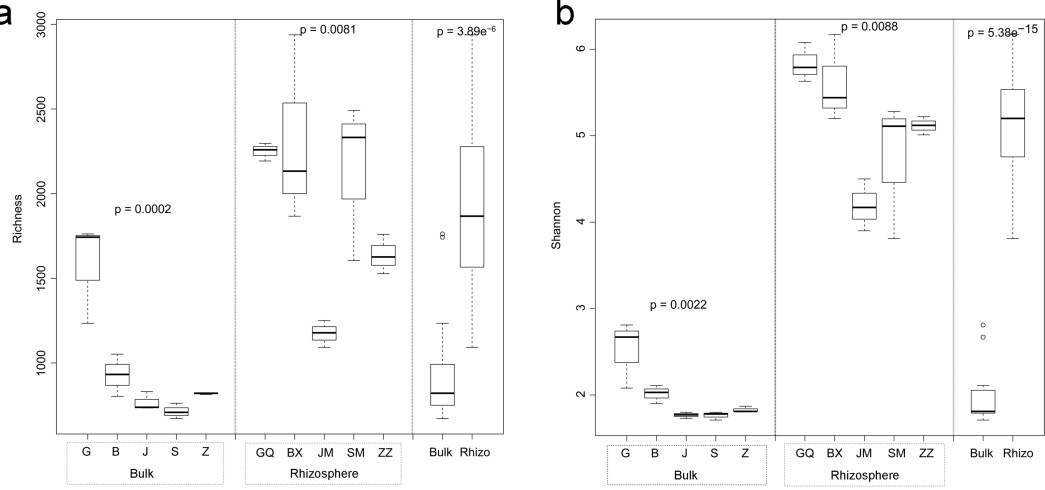

**Figure 1** **Multiple comparisons of OTU richness (A) and α-diversity (Shannon's index) (B) between rhizospheric and bulk soil bacterial communities, and among plant species within soil type.** Within boxes, horizontal bars indicate medians, while the tops and bottoms of boxes illustrate 75th and 25th quartiles, respectively. Small circles represent outliers in bulk samples. G, B, J, S, Z indicate bulk soil samples from *Lycium ruthenicum*, *Limonium gmelinii*, *Halocnemum strobilaceum*, *Halostachys caspica*, and *Kalidium foliatum*, respectively, while GQ, BX, JM, SM and ZZ denote the corresponding rhizosphere samples from each species. Significance levels (*p* values) are provided for among-species comparisons within soil type (bulk versus rhizosphere), as well as between soil types.

samples (*P* < 0.01; Fig. 1). The number of OTUs identified in the rhizosphere of each species ranged from a minimum of 2,342 (for *Halocnemum strobilaceum*) to a maximum of 4,602 (*Limonium gmelinii*); within species, the number of OTUs present in all three replicates ranged from 317 to 729. In bulk soil samples, the number of OTUs detected for each species ranged from 1,108 to 3,688, and between 94 and 220 OTUs were common to all three replicates within species (Table S1).

Bacterial community diversity was higher in the rhizosphere versus bulk soil samples (ANOVA *P* < 0.01) (Fig. 1). Rhizobacterial diversity did not differ among *Halocnemum strobilaceum*, *Halostachys caspica* and *Kalidium foliatum*, but was lower in these species than in *Limonium gmelinii* and *Lycium ruthenicum* (*P* < 0.05). Diversity in *Lycium ruthenicum* bulk soil samples was higher than in the other four plant species (*P* < 0.05). There were 1005, 1001, 677, 589 and 510 OTUs exclusive to the rhizobacterial communities associated with *Halostachys caspica*, *Limonium gmelinii*, *Lycium ruthenicum*, *Kalidium foliatum*, and *Halocnemum strobilaceum*, respectively; of these, 135, 187, 242, 88 and 63 OTUs, respectively, were found in all within-species replicates. Comparing across species, 242 OTUs were identified in all bulk soil samples and 647 OTUs in all rhizosphere samples; only 31 and 87 of these OTUs, respectively, were found in all three individuals of each species (Fig. 2). Abundant OTUs (i.e., those accounting for >0.1% of sequences) common to the rhizospheres of all halophytes belonged to 16 genera, including *Acinetobacter*, *Aliifodinibius*, *Citrobacter*, *Deferrisoma*, *Exiguobacterium*, *Gracilimonas*, *Halomonas*, *Marinobacter*, *Pseudomonas*, *Thioprofundum* and others. Considering bulk soils, the most

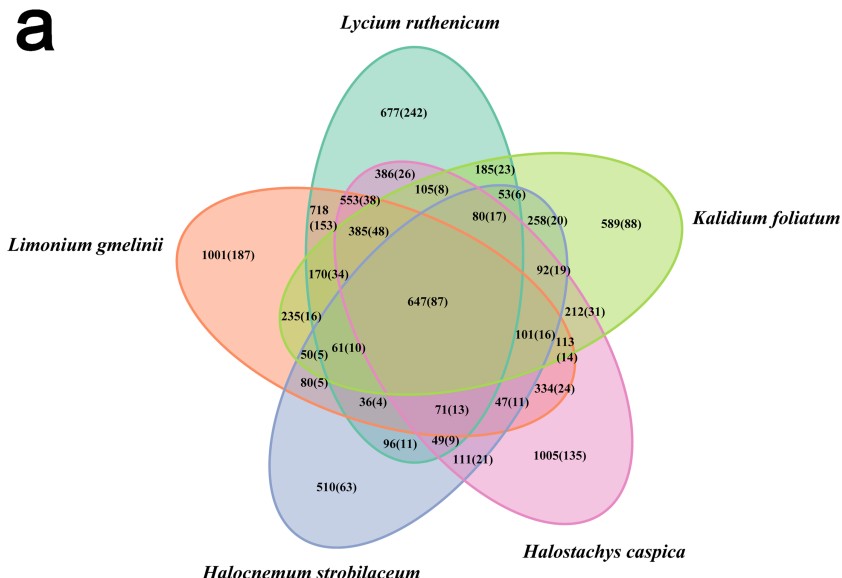

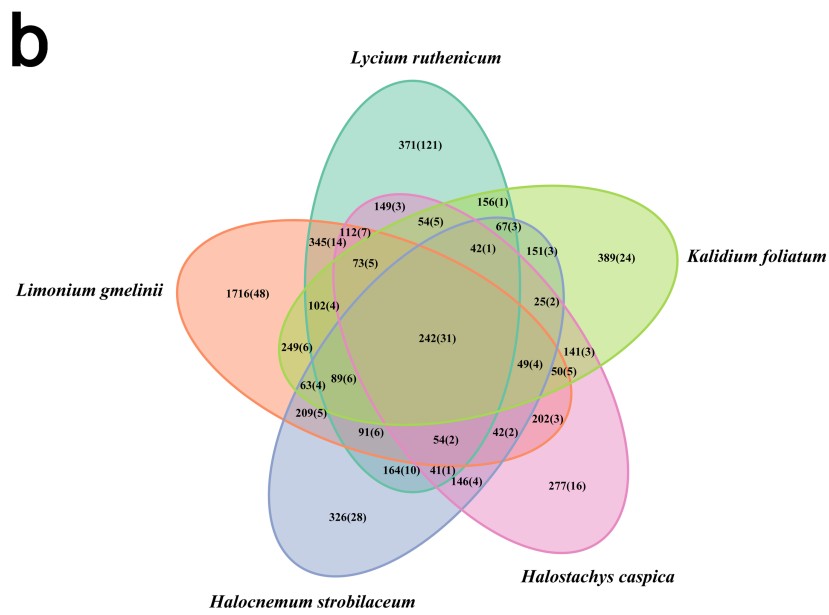

**Figure 2   Venn diagram showing the OTUs shared among different samples.** (A) Rhizosphere samples, (B) bulk soil samples. Information on OTU numbers is provided as follows: the total number of OTUs detected across all three replicates for each species (the number of OTUs shared among the three replicates).

abundant OTUs came from four genera: *Acinetobacter*, *Citrobacter*, *Exiguobacterium* and *Pseudomonas* (Fig. S4).

## Bacterial community structure

A total of 36 phyla, 61 classes, 201 families and 617 genera were identified over all samples. In the bulk soil samples, Proteobacteria and Firmicutes were the dominant phyla.

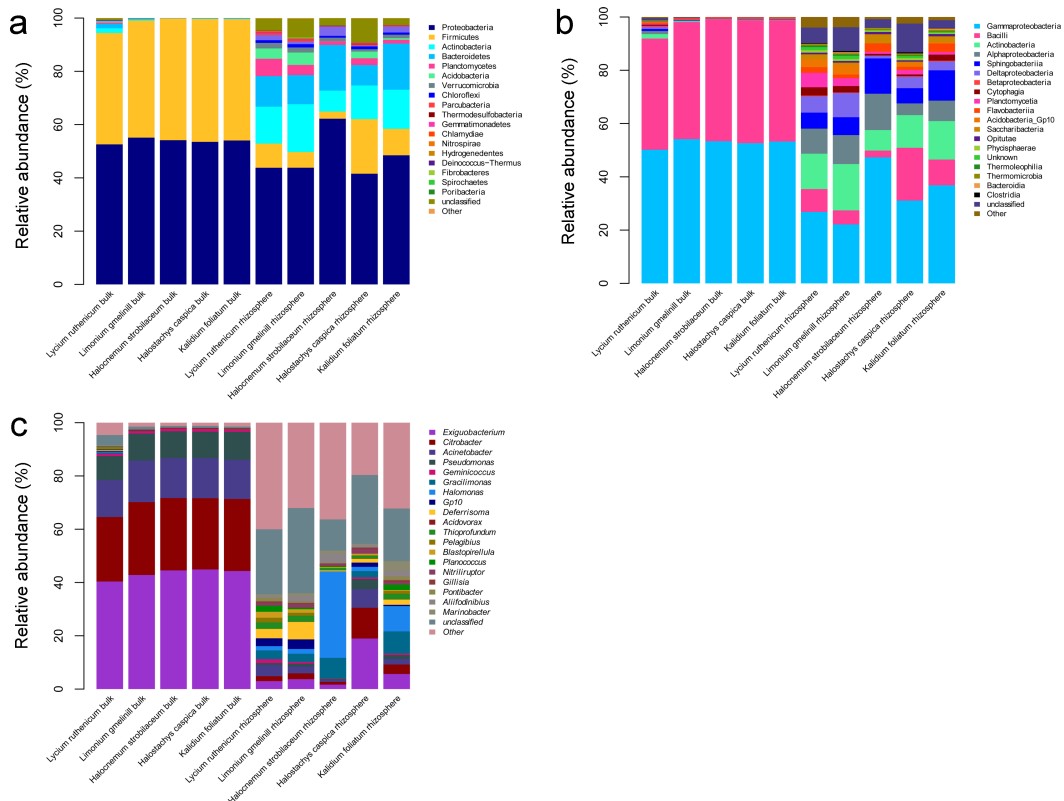

**Figure 3** Diagrams depicting community composition and relative abundance at different taxonomic level. (A) phylum, (B) class, (C) genus.

Gammaproteobacteria and Bacilli were the dominant classes, and *Acinetobacter*, *Bacillus*, *Citrobacter*, *Exiguobacterium* and *Pseudomonas* were the most abundant genera (Fig. 3). In the rhizosphere samples, Acidobacteria, Actinobacteria, Bacteroidetes, Chloroflexi, Firmicutes, Planctomycetes, Proteobacteria and Verrucomicrobia were the most abundant phyla. Actinobacteria, Bacilli, Cytophagia, Flavobacteriia, Planctomycetia, Sphingobacteriia, Alphaproteobacteria, Deltaproteobacteria and Gammaproteobacteria were the most abundant classes. The most common genera included *Acinetobacter*, *Aliifodinibius*, *Citrobacter*, *Deferrisoma*, *Exiguobacterium*, *Geminicoccus*, *Gp10*, *Gracilimonas*, *Halomonas*, *Marinobacter*, *Pseudomonas* and *Thioprofundum* (Fig. 3).

## Differences between rhizosphere and bulk soil samples

The most abundant bacterial groups in bulk soil communities were less common in rhizosphere communities, whereas some low abundance groups in the former were more common in the latter. At the phylum level, the Firmicutes and Proteobacteria was relatively less abundant in the rhizosphere, especially in the case of Firmicutes ($P < 0.001$), whereas the Acidobacteria, Actinobacteria, Bacteroidetes, Chloroflexi, Planctomycetes and Verrucomicrobia were more abundant. A similar pattern was also observed at lower

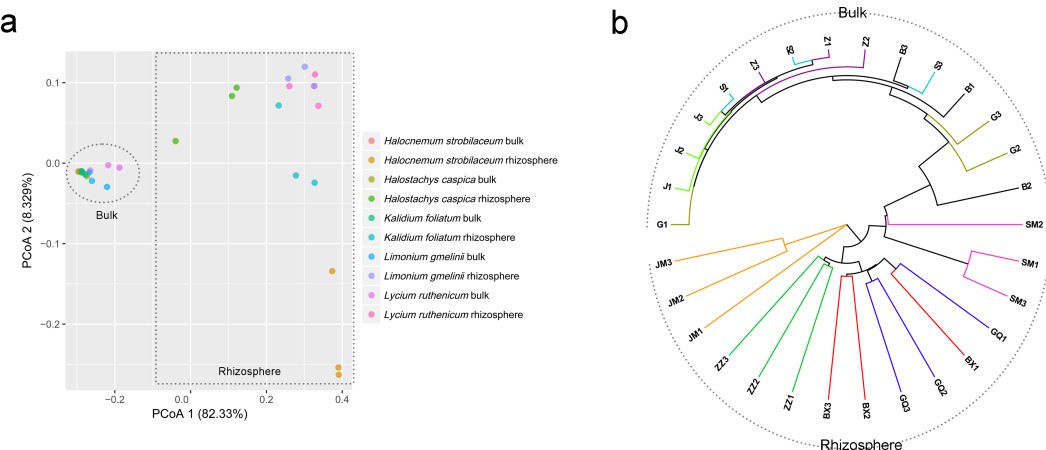

**Figure 4** **Principal coordinate analysis (PCoA) (A) and UPGMA clustering diagram (B) of soil samples.** G, B, J, S and Z represent bulk soil samples from *Lycium ruthenicum*, *Limonium gmelinii*, *Halocnemum strobilaceum*, *Halostachys caspica* and *Kalidium foliatum*, respectively, while GQ, BX, JM, SM and ZZ represent rhizosphere samples associated with these species.

taxonomic ranks; however, overall, there were more abundant groups in the rhizosphere. Comparing bulk soil, the following genera were lower in abundance: *Acinetobacter*, *Citrobacter*, *Exiguobacterium*, *Halomonas* and *Pseudomonas*. Meanwhile, *Aciditerrimonas*, *Aliifodinibius*, *Deferrisoma*, *Fodinicurvata*, *Geminicoccus*, *Gp10*, *Gracilimonas*, *Marinobacter*, *Nitriliruptor*, *Planococcus* and *Thioprofundum* were more abundant in the rhizosphere samples ($P < 0.01$) (Fig. 3).

Community composition differed significantly between rhizosphere and bulk soil samples (ANOSIM, $R = 0.961$, $P = 0.001$). A PCoA analysis illustrates these differences in community structure (Fig. 4A). Rhizosphere and bulk soil samples are separated on the first PCoA axis, which explained 82.33% of the variance. Bulk soil samples clustered together, indicating a high degree of similarity among their bacterial communities. However, rhizosphere samples were less tightly aggregated than bulk soil samples. Consistent with the PCoA results, the UPGMA tree also distinguished rhizosphere communities from bulk soil communities (Fig. 4B).

## Community structure differences among the five halophytes

Significant variation in community structure was observed among the five species' rhizospheres (Adonis $R^2 = 0.703$, $P = 0.001$). However, pairwise differences between each species pair, though relatively large, were not significant ($P > 0.05$) (Table 2). The rhizobacterial community of *Lycium ruthenicum* was most similar to that of *Limonium gmelinii*. Meanwhile, the *Halocnemum strobilaceum* rhizobacterial community was most dissimilar to those of other species. As determined by a LEfSe analysis, rhizobacterial community composition differed among the five halophytes (Fig. 5). The genera *Gimesia* and *Pelagibius* were significantly more abundant in the *Lycium ruthenicum* rhizosphere; *Albidovulum*, *Bauldia*, *Deferrisoma*, *Geminicoccus*, *Gp10* and *Thiohalomonas* were more

**Table 2  Adonis analysis of bacterial community composition for bulk soil and rhizosphere samples.**

|  | *Lycium ruthenicum* | *Limonium gmelinii* | *Halocnemum strobilaceum* | *Halostachys caspica* | *Kalidium foliatum* |
|---|---|---|---|---|---|
| *Lycium ruthenicum* |  | 0. 323 (0.3) | 0.399 (0.199) | 0.319 (0.289) | 0.367 (0.089) |
| *Limonium gmelinii* | 0. 324 (0.108) |  | 0.145 (0.486) | 0.042 (0.894) | 0.086 (0.615) |
| *Halocnemum strobilaceum* | 0.697 (0.126) | 0.718 (0.098) |  | 0.096 (0.605) | 0.175 (0.499) |
| *Halostachys caspica* | 0.528 (0.104) | 0.513 (0.11) | 0.750 (0.104) |  | 0.016 (0.904) |
| *Kalidium foliatum* | 0.457 (0.097) | 0.504 (0.102) | 0.521 (0.098) | 0.574 (0.114) |  |

**Notes.**
Data was shown in format of $R_2$ ($P$ value); values below diagonal are for rhizosphere samples, and above diagonal for bulk samples.

common in the *Limonium gmelinii* rhizosphere, while *Citrobacter*, *Exiguobacterium* and *Pseudomonas* in the *Halostachys caspica* rhizosphere, *Gracilimonas*, *Jiangella*, *Marinimicrobium*, *Planococcus* and *Pontibacter* in the *Kalidium foliatum* rhizosphere, and *Fodinicurvata*, *Halomonas*, *Mesorhizobium* and *Salegentibacter* in the *Halocnemum strobilaceum* rhizosphere.

In contrast, an Adonis analysis found no significant differences in community composition among bulk soil samples from different species ($R^2 = 0.300$, $P = 0.413$). However, LEfSe analyses found differences in composition in bulk soil communities for the five species (Fig. 6). The bacterial community associated with *Lycium ruthenicum* was relatively distinct from that of other species, mainly due to differences in the abundance of the following families: Acidimicrobiaceae, Anaerolineaceae, Alteromonadaceae, Bacillaceae, Chromatiaceae, Demequinaceae, Planctomycetaceae, Puniceicoccaceae, Rhodobiaceae and Sprospiraceae. Meanwhile, *Kalidium foliatum* communities were indistinguishable from those of the other four species.

## Correlations between bacterial diversity, community structure and soil properties

Relationships between microbial community diversity and structure, and soil bio-chemical properties were assessed with Pearson correlation coefficients and canonical correspondence analysis (CCA). The soil TOC, TON and AP were all strongly positively correlated, while EC was negatively correlated with all other soil variables except pH. Bacterial community diversity and richness were positively correlated with soil TON and AP, but negatively correlated with EC (Table 3). The first canonical axis (CCA1) was negatively correlated with EC, while the second canonical axis (CCA2) was positively correlated with SWC, but negatively correlated with EC. The SWC, TOC, SOM, TON and AP were more important in determining the bacterial community composition (as represented by longer arrows) than pH and EC, indicating that EC played only a minor role in the shaping of community structure. Rhizosphere communities from *Lycium ruthenicum* and *Limonium gmelinii* samples were positively associated with higher SWC, TON and AP. Meanwhile, *Halostachys caspica* rhizosphere communities were positively correlated with EC, and *Halocnemum strobilaceum* communities negatively correlated (Fig. 7A). Considering the bulk soil samples, CCA1 was positively correlated with EC and
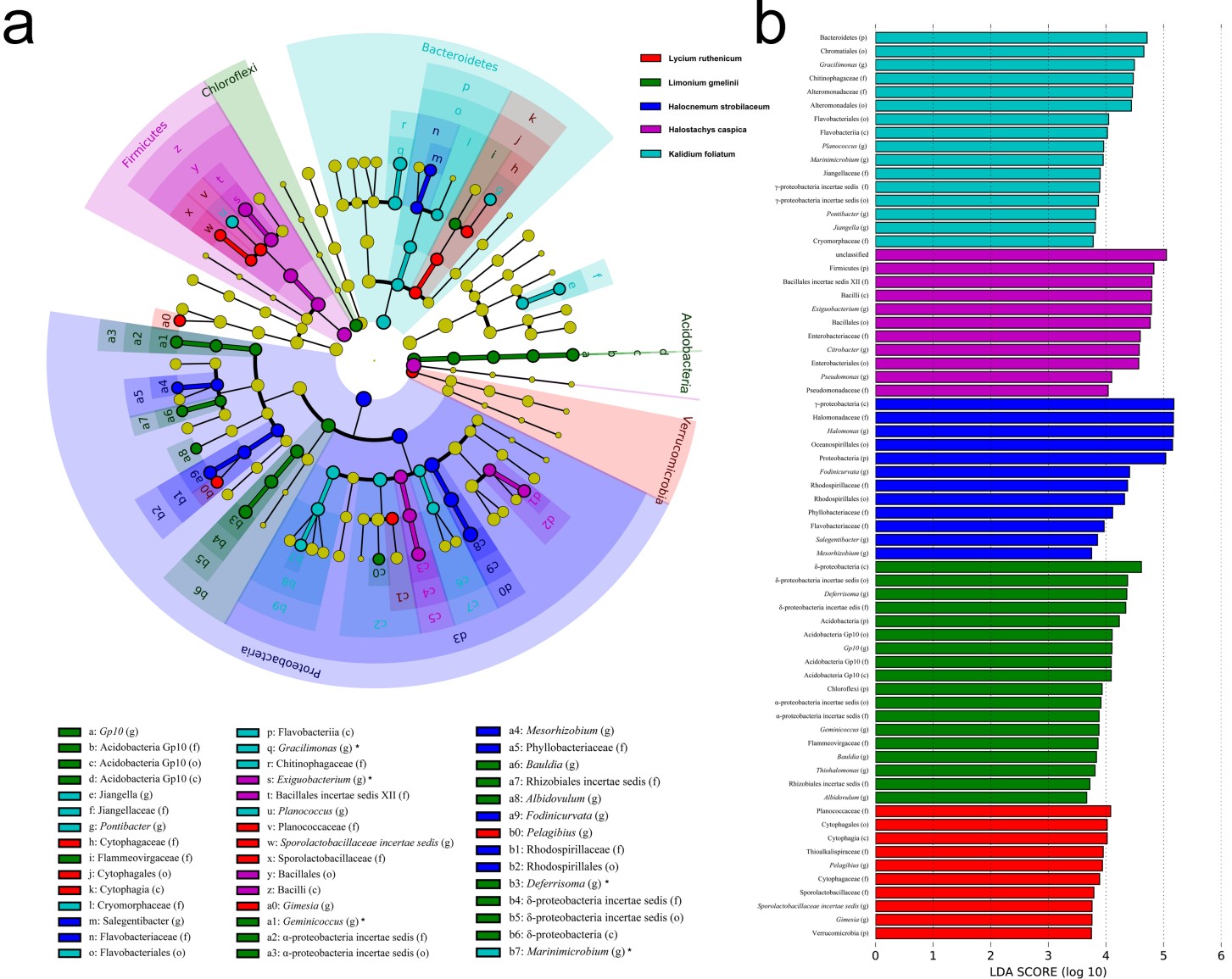

**Figure 5   LEfSe analysis at multiple taxonomic levels comparing rhizobacterial community composition for five focal plant species.** (A) Clado-gram illustrating the taxonomic groups explaining the most variation among rhizobacterial communities. Each ring represents a taxonomic level, with phylum (p), class (c), order (o), family (f) and genus (g) emanating from the center to the periphery. Each circle is a taxonomic unit found in the dataset, with circles or nodes shown in colors (other than yellow) indicating where a taxon was significantly more abundant. (B) Histogram of the LDA scores computed for groups with differential abundance among the rhizobacterial communities of the five plant species.

negatively correlated with the TOC, TON and pH. The second axis (CCA2) was positively correlated with AP and SWC. Apart from *Lycium ruthenicum,* the other species' samples were negatively correlated with SOM, TON and AP, but positively correlated with EC. The TON, AP and pH had a stronger influence on community structure than SOM or EC (Fig. 7B).

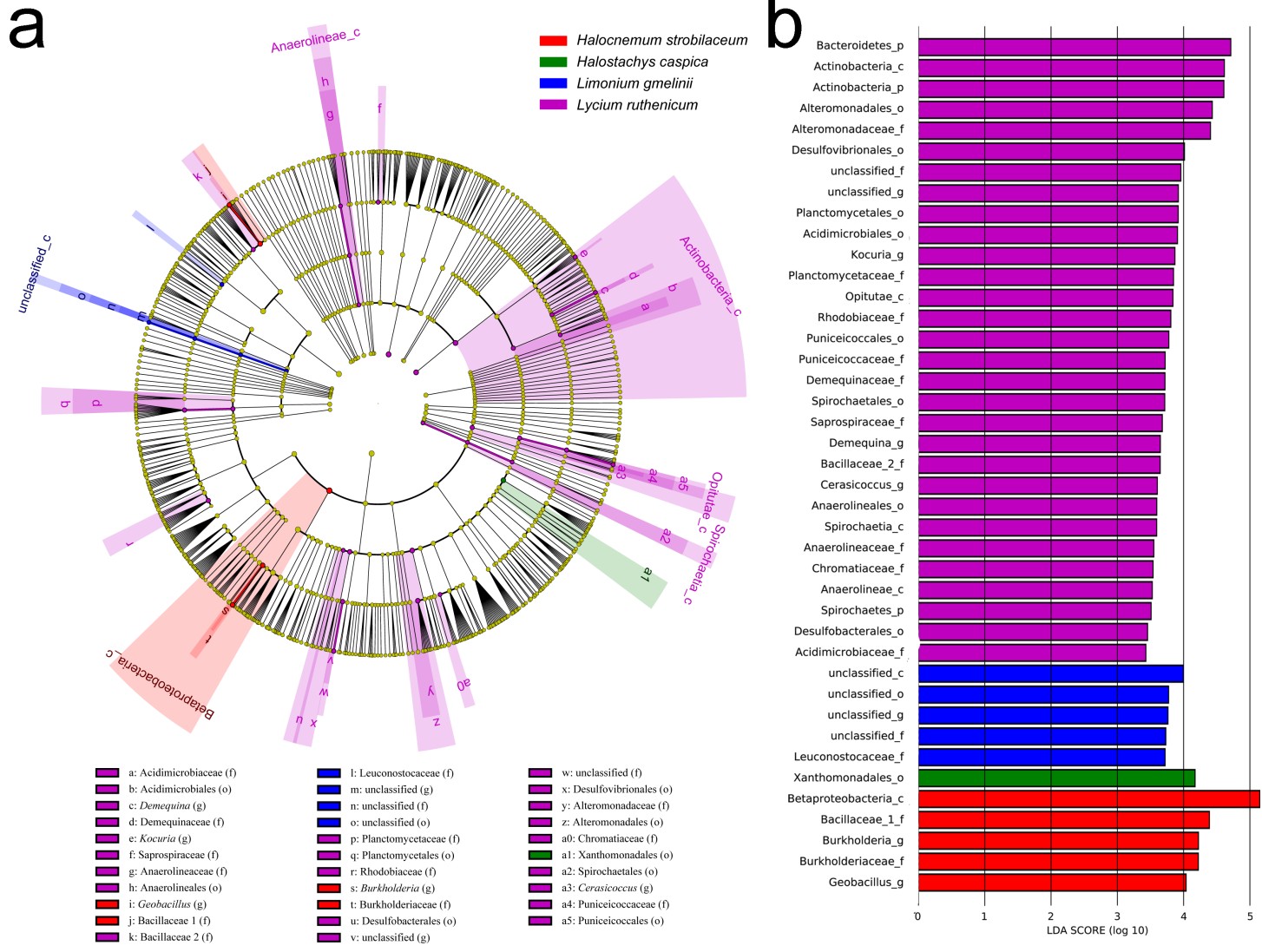

**Figure 6** **LEfSe analysis of bacterial community composition for bulk soil samples from four plant species.** (A) Cladogram illustrating the taxonomic groups that explain the most variation among the bacterial communities. (B) Histogram of the LDA scores computed for groups with differential abundance among the bacterial communities of the four plant species. *Kalidium foliatum*-associated communities did not form a separate group (from other plant species' associated communities) and, therefore, these are absent from the cladogram and histogram.

## DISCUSSION

### Bacterial community structure in saline soils as compared to other environments

Soil salinity has important effects on the distribution of plant communities, their composition and diversity (*Xi et al., 2016*), and may also affect soil bacterial diversity and community structure (*Fang et al., 2016*; *Pavloudi et al., 2016*). Plant communities in salinized habitats are dominated by halophytes, and the abundance and diversity of associated microbial communities, in saline or hypersaline terrestrial environments,

**Table 3  Pearson correlation coefficients among soil chemical properties, and between soil properties and community diversity.**

|  | TOC | TON | AP | SWC | pH | EC | OTU richness | Shannon index |
|---|---|---|---|---|---|---|---|---|
| TOC |  |  |  |  |  |  | 0.446 | 0.484 |
| TON | 0.970*** |  |  |  |  |  | 0.549* | 0.620* |
| AP | 0.728** | 0.761*** |  |  |  |  | 0.559* | 0.609* |
| SWC | 0.479* | 0.476* | 0.137 |  |  |  | 0.485 | 0.435 |
| pH | 0.420 | 0.467 | 0.151 | 0.301 |  |  | 0.504 | 0.459 |
| EC | −0.211 | −0.235 | −0.126 | −0.206 | 0.235 |  | −0.093 | −0.054 |

∗ indicates significance level, * $P < 0.05$, ** $P < 0.01$, *** $P < 0.001$.

EC, electrical conductivity; TOC, total organic carbon; SOM, soil organic matter; TON, total nitrogen; AP, available phosphorous; SWC, soil water content..

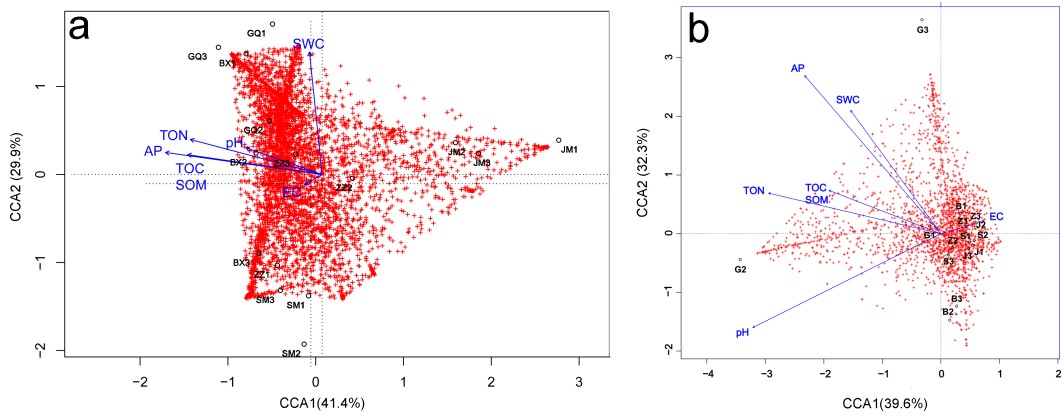

**Figure 7  Canonical correspondence analysis (CCA) of the effect of soil properties on bacterial community structure in the rhizosphere (A) and bulk soil samples (B).** The soil properties are indicated with arrows, and include soil pH, electrical conductivity (EC), total organic carbon (TOC), soil organic matter (SOM), total nitrogen (TON), phosphorous (AP) and soil water content (SWC). The percentage of variation explained by each axis is provided. GQ, BX, JM, SM and ZZ represent rhizosphere samples associated with *Lycium ruthenicum*, *Limonium gmelinii*, *Halocnemum strobilaceum*, *Halostachys caspica* and *Kalidium foliatum*, respectively. G, B, J, S and Z represent bulk soil samples from *Lycium ruthenicum*, *Limonium gmelinii*, *Halocnemum strobilaceum*, *Halostachys caspica* and *Kalidium foliatum*, respectively.

is usually low (*Takekawa et al., 2006*; *Jiang et al., 2007*; *Foti et al., 2008*). In this study, bacterial communities were very low in richness and diversity compared to documented communities in forests, grasslands and agricultural areas (*Rampelotto et al., 2013*), maize crop soils (*Garcia-Salamanca et al., 2013*) and even other saline soils (*Canfora et al., 2014*); however, metrics were consistent with those found in other "extreme" hypersaline soils in semiarid Mediterranean regions (*Canfora et al., 2015*). The soils surrounding *Halocnemum strobilaceum* had the lowest diversity and richness, perhaps as a result of the relatively high soil EC and low nutrient availability. Soils associated with *Lycium ruthenicum* were the most diverse of all studied plant species. Considering that nutrient availability in *Lycium ruthenicum*-associated soils was only mediocre, other factors (i.e., plant community composition) likely affected the soil bacterial community (*Ravit, Ehrenfeld & Haggblom, 2003*; *Cao et al., 2014*). Indeed, based on field observations, the

plant community surrounding focal *Lycium ruthenicum* individuals was more diverse and had higher percent cover than for other species.

   Bulk soil samples had similar, relatively simple bacterial communities. Only four genera, *Acinetobacter*, *Citrobacter*, *Exiguobacterium* and *Pseudomonas*, were common, which differs from the bacterial communities of saline soils in Inner Mongolia (*Borruso et al., 2014*) and the Shandong Peninsula coast (*Tian & Zhang, 2017*). Differences in community composition may be due to distinct climatic and/or soil properties among geographic regions (*Ben-David et al., 2011*). However, the dominance of Bacilli (Firmicutes) and $\gamma$-proteobacteria in the study region is consistent with previous studies (*Tang et al., 2011*; *Borsodi et al., 2013*), confirming that these two taxa are important in saline or hypersaline environments. A low level of structural differentiation was also found when comparing communities associated with different plant species, as determined by a LEfSe analysis. The highest differentiation was observed between *Lycium ruthenicum* and the other species; in contrast, *Kalidium foliatum* communities were indistinguishable from those of other species. This suggests that soil bacterial communities are highly similar in arid environments, at least on a small geographic scale. Microhabitat similarities may produce such similar bacterial communities, as low plant cover and a lack of plant litter are common to arid environments.

## Bacterial community composition in the rhizosphere versus bulk soils

Rhizosphere effects may be an important driving force shaping microbial communities and leading to compositional differences between the rhizosphere and the bulk soils (*Morgan & Whipps, 2001*). In this study, bacterial communities differed between the rhizosphere and bulk soils, with higher community diversity and richness in the rhizosphere, consistent with previous studies (*Avis et al., 2008*; *Borruso et al., 2014*; *Edwards et al., 2015*; *Chaudhary et al., 2017*; *Yang et al., 2017*). Higher diversity may be a result of root exudates, which can raise nutrient concentrations (*Li et al., 2014*). While soil properties were not measured exhaustively here, the soil TOC and TON contents estimated in the rhizosphere of *Limonium gmelinii* and *Lycium ruthenicum* were approximately 5–7 times higher than in the bulk soils. However, it should be noted that bulk soils may have greater microbial richness than the rhizosphere in some cases (*Shange et al., 2012*; *Carbonetto et al., 2014*; *Gomes et al., 2014*; *Tian & Zhang, 2017*). These divergent results suggest that the soil type and plant species identity have complex effects on bacterial communities, with the strength of their effects depending on abiotic and biotic conditions (*Berg & Smalla, 2009*).

   In addition to diversity differences, the composition of bacterial communities also differed between the rhizosphere and bulk soil samples, as revealed by PCoA and cluster analysis. Compared to bulk soils, the rhizosphere communities had more groups with relative abundance >1%, such as the following phyla: Actinobacteria, Bacteroidetes, Firmicutes, Planctomycetes and Proteobacteria. Dominance by a greater number of groups has also been reported in other saline ecosystems, both marine and terrestrial (*Tang et al., 2011*; *Marasco et al., 2013*; *Oliveira et al., 2014*; *Soussi et al., 2016*). At the genus level, many genera were abundant in the rhizosphere, for example *Acinetobacter*, *Bacillus*, *Citrobacter*, *Deferrisoma*, *Exiguobacterium*, *Haliea*, *Halomonas*, *Marinimicrobium*, *Marinobacter*,

*Methylohalomonas*, *Microbulbifer*, *Planococcus*, *Pseudomonas* and *Thioprofundum* (Fig. 3, Fig. S5). Meanwhile, the richness of groups common in bulk soils (i.e., γ-proteobacteria and Firmicutes) was reduced in the rhizosphere; for example, the abundance of Firmicutes was about 80% lower. The low abundance of Firmicutes in the rhizosphere has been reported many times; see one such case study in barley, where Firmicutes is almost excluded from the rhizosphere (*Bulgarelli et al., 2015*). Although γ-proteobacteria were less abundant in the rhizosphere, as observed by a decrease in the number of occurrences of *Acinetobacter*, *Citrobacter* and *Pseudomonas* species, γ-proteobacteria remained the most abundant class, as found in many plant-associated bacterial communities (*Mukhtar et al., 2017*).

Shifts in bacterial community composition in bulk soils versus the rhizosphere may be the consequence of active selection by plants (*Kowalchuk et al., 2002*). As many endophytes and bacteria colonizing root surfaces have beneficial effects, such as nitrogen fixation, phytohormone production, nutrient supply and pathogen suppression (*Rosenblueth & Martinez-Romero, 2006*; *Hardoim, van Overbeek & van Elsas, 2008*), they typically promote plant growth and can alleviate salt stress in halophytes (*Ali et al., 2015*). Some *Microbulbifer* and *Planococcus* species have the ability to degrade complex hydrocarbons (*See-Too et al., 2017*). Meanwhile, *Bacillus*, *Exiguobacterium*, *Halomonas*, *Planococcus* and *Pseudomonas* can generate 1-aminocyclopropane-1-carboxylic acid (ACC) deaminase to convert the ethylene precursor ACC into ammonia and α-ketobutyrate; this has the effect of lowering the ethylene concentration within plant tissues, reducing its constraining effect on root elongation and general plant growth (*Siddikee et al., 2010*). Moreover, some *Bacillus*, *Halomonas* and *Pseudomonas* species can produce indole-3-acetic acid (IAA) to confer salt tolerance (*Tiwari et al., 2011*). *Halomonas* species are able to tolerate a wide set of abiotic stresses, and promote plant growth via IAA production, phosphate solubilization, nitrogen fixation and more (*Mapelli et al., 2013*). In rhizobacterial communities, *Bacillus* and *Pseudomonas* species are critically important for plant growth; these species have multiple functional activities, including phosphate solubilization, phytopathogen inhibition (*Prashar, Kapoor & Sachdeva, 2014*) and auxin production. Why these species are less abundant in the rhizosphere (versus bulk soils) remains unclear. It could be caused by competition among rhizobacteria, but this hypothesis needs verification. Moreover, the effects of many other common genera in the rhizosphere (whether beneficial, harmful or neutral), as well their inter-relationships (mutualistic or competitive) with plant growth promoting species, need further investigation.

## Differences among halophytes in rhizobacterial communities and relationship to soil properties

A number of studies have demonstrated that microbial community composition is plant species specific (*Andreote et al., 2009*; *Poli et al., 2016*), leading to unique rhizobacterial communities among species. Here, five halophyte-associated rhizobacterial communities varied both in diversity (ANOVA $P < 0.01$) and structure (Adonis $R^2 = 0.703$, $P = 0.001$). Rhizobacterial diversity was lower in *Halostachys caspica*, *Halocnemum strobilaceum* and *Kalidium foliatum* than in *Limonium gmelinii* and *Lycium ruthenicum* ($P < 0.05$). While community composition also seemed to vary, differences were not significant (Table 2).

*Halomonas* was the dominant genera in *Halocnemum strobilaceum* communities (32.4% of total abundance), a significantly higher proportion than in the other plant-associated communities; this trend is consistent with previous studies (*Al-Mailem et al., 2010*; *Marasco et al., 2016*), including of other halophytes (*Borsodi et al., 2015*). In the *Halostachys caspica* rhizosphere, *Exiguobacterium* was the most common genus, followed by *Citrobacter*, *Acinetobacter* and *Pseudomonas*. Meanwhile, in the *Kalidium foliatum* rhizosphere, *Halomonas*, *Exiguobacterium* and *Gracilimonas* were most abundant. The rhizobacterial communities of *Limonium gmelinii* and *Lycium ruthenicum* were highly similar (Adonis $R^2 = 0.324$, $P = 0.108$), perhaps as a result of similar soil properties. The relatively high abundance of *Exiguobacterium* in the rhizosphere is unique to this study, suggesting that it is potentially important in the study region, but this requires further investigation.

Furthermore, among replicates for a given plant species, only a small proportion of OTUs occurred in all three. Interestingly, these shared OTUs were usually highly abundant overall, whereas OTUs that were found only in one or two individuals were mostly rare OTUs (with a relative abundance less than five). Thus, bacterial communities may have individual specificity, potentially influenced by a plant's genotype, age or health (*Zhu et al., 2013*; *Belimov et al., 2015*). The OTUs found across all five plant species belong to 16 genera in total, and some, such as *Deferrisoma*, *Exiguobacterium*, *Geminicoccus*, *Gracilimonas* and *Marinimicrobium*, were also included in the taxa identified in the LEfSe. As successful colonizers of halophyte plants in saline-arid lands, these may be good plant growth promotion rhizobacteria (PGPR) candidates.

Soil properties are also important determinants of soil microbial communities (*Szymanska et al., 2016a*; *Rathore, Chaudhary & Jha, 2017*). In particular, soil salinity is considered a primary environmental factor, directly or indirectly driving the composition and diversity of prokaryotic communities (*Marasco et al., 2016*; *Zhong et al., 2016*). In this study, soil EC had a positive influences on the bulk soil bacterial communities, and other soil factors also shaped bacterial community composition. Soil nutrients (TON, TOC and AP) were most important for *Lycium ruthenicum*. Similarly, a recent study found that variation in prokaryotic community structure was significantly correlated with the TON and $PO_4^{3-}$ concentration (*Zhong et al., 2016*). In the rhizosphere, EC was negatively correlated with rhizobacterial community diversity and structure, though its influence on community structure was only minor as revealed by the CCA. In contrast, the effects of soil nutrients, such as TOC, SOM, TON and AP, were much stronger than those of EC or soil pH. The low diversity and compositional variation seen in *Halostachys caspica* and *Halocnemum strobilaceum* rhizobacterial communities were negatively correlated with soil TOC, SOM, TON and AP. These results imply that, when salinity levels are similar among habitats, the influence of salinity on the rhizosphere microbial community structure is relatively weak. In these cases, rhizobacterial community structure is more strongly influenced by the host plant identity and soil nutrient content, whereas salinity has a minor effect.

Overall, variation in rhizobacterial community structure among the five study plant species was significant, though pairwise differences did not reach significance despite the large number of OTUs that were unique to each species. This may be explained by the relatively low total abundance of rare OTUs, as well as the low proportion of shared OTUs
in a given plant species. While abundant OTUs were shared across all five plant species (and accounted for a large proportion of total richness in each), they differed in relative abundance. This suggests that, in similar saline habitats, plants tend to select similar bacterial species to colonize roots, perhaps as a consequence of adaptation to salinity stress; these species may act to promote plant growth or alleviate salt stress. Coevolution between microorganisms and associated plant species has been addressed in leguminous *Rhizobium* species (*Wang et al., 2018*). Community differences (especially in the richness of abundant groups) may be closely related to differences in root exudates or plant litter between plant species (*Chaudhary et al., 2015*), as supported by the results here. However, the effects of other factors, such as ion concentration, and interactions between microbiomes within a community (*Chen et al., 2013*; *Poosakkannu, Nissinen & Kytoviita, 2017*) cannot be excluded.

## CONCLUSIONS

The present study investigated the composition and diversity of rhizobacterial communities in five co-occurring halophytic species growing in salinized, arid desert soils within the Ebinur Lake Wetland Reserve in Northwestern China. Significant differences were found between the rhizosphere and bulk soil communities, both in diversity and bacterial composition. Diversity was higher in the rhizosphere than in the bulk soils. Abundant taxonomic groups (from phylum to genus) in the rhizosphere were much more diverse than in bulk soils. Actinobacteria, Bacteroidetes, Firmicutes, Planctomycetes and Proteobacteria were the most abundant phyla in the rhizosphere, while Firmicutes and Proteobacteria were common in bulk soils. Comparing among species, significant differences in rhizobacterial diversity and identity were observed. The diversity of *Halostachys caspica*, *Halocnemum strobilaceum* and *Kalidium foliatum* associated communities was lower than that of *Limonium gmelinii* and *Lycium ruthenicum* communities. Furthermore, the composition of *Halostachys caspica* and *Halocnemum strobilaceum* communities was very different from that of *Limonium gmelinii* and *Lycium ruthenicum* communities. Thus, plant species identity can have important effects on root-associated bacterial communities. Diversity was positively correlated with soil nutrients, including TOC, SOM, TON and AP, but negatively correlated with EC, though the effects of EC were much lower than those of soil nutrient content. In conclusion, halophytic plant species played an important role in shaping associated rhizosphere bacterial communities. Furthermore, when salinity levels were constant, soil nutrients emerged as key factors structuring bacterial communities. These results provide insight into the nature of halophyte microbial communities in arid regions, as well as the factors shaping these communities. However, pairwise differences among the five rhizobacterial communities were not significant, despite some evidence for differentiation among plant species. Further studies involving more halophyte species and individuals per species are necessary to elucidate plant species identity effects on the rhizosphere for co-occurring halophytes.

## ACKNOWLEDGEMENTS

We would like to thank LetPub, Elizabeth Tokarz at the Yale University and Dr. Emily Drummond at the University of British Columbia for their assistance with English language and grammatical editing of the manuscript. We would also like to thank Dr. Rodrigo Taketani and the anonymous reviewers for their comments and suggestions that improved the quality of our manuscript.

### Funding

This work was supported by the National Science Foundation for Post-doctoral Scientists of China (grant no: 2016M592866), the National Natural Science Foundation of China (grant no. 31560131 and 31500309), and the Scientific Research Fund for Doctors of Xinjiang University (BS150259). The funders had no role in study design, data collection and analysis, decision to publish, or preparation of the manuscript.

### Grant Disclosures

The following grant information was disclosed by the authors:
National Science Foundation for Post-doctoral Scientists of China: 2016M592866.
National Natural Science Foundation of China: 31560131, 31500309.
Scientific Research Fund for Doctors of Xinjiang University: BS150259.

### Competing Interests

The authors declare there are no competing interests.

### Author Contributions

- Yan Li conceived and designed the experiments, prepared figures and/or tables, authored or reviewed drafts of the paper, approved the final draft.
- Yan Kong analyzed the data, approved the final draft.
- Dexiong Teng analyzed the data, prepared figures and/or tables, approved the final draft.
- Xueni Zhang authored or reviewed drafts of the paper, approved the final draft.
- Xuemin He contributed reagents/materials/analysis tools, approved the final draft.
- Yang Zhang performed the experiments, approved the final draft.
- Guanghui Lv conceived and designed the experiments, authored or reviewed drafts of the paper, approved the final draft.

### Data Availability

NCBI Sequence Read Archive: SRP129060.

### Supplemental Information

Supplemental information for this article can be found online at http://dx.doi.org/10.7717/peerj.5508#supplemental-information.

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
