# Peer review of "Rhizobacterial communities of five co-occurring desert halophytes"

_PeerJ, doi:10.7717/peerj.5508_

## Round 0.1 · original submission · Major Revisions

Dear authors,

Your ms. has been evaluated by three reviewers. All reviewers recognize that your ms. has merits. But they also raised major concerns (in particular reviewers 2 & 3). These deficiencies will have to be carefully addressed to make the ms. acceptable. In particular:

INTRODUCTION:
* Clearly state/identify specific knowledge gap(s) addressed by this work.
* additional papers to be cited

MATERIAL & METHODS:
* The methodology is not detailed enough (reviewers 1 -incl annotated ms; 2; & 3). Please complete all missing information (including sampling methodology and map; method used to separate rhizosphere from roots; procedure and pipeline to analyse sequences and OTUS ; method to collect bulk soil ; methods used for soil chemical analyses ; etc.)

RESULT INTERPRETATION
* Please clarify what do the OTU numbers mean in the Ven diagram (see reviewer 3)?
* A ven diagram could also include bacterial communities of the bulk soil to analyse the selective process by plant species (rev 2)
* In addition to the NMDS visualisation, ANOSIM-like tests are needed to assess plant species effect on community composition
Linked to this: Only 3 replicates were used : please evaluate whether this number is sufficient to test the effect of plant species.
* it is needed to test for possible correlation between the communities and chemical analysis (salt concentration ...) using adequate multivariate analyses. See the suggestions of reviewers 2 & 3 about the way you should/could test the possible effect of salinity. All the results should be clearly presented in the Results section

TABLES & FIGURES
* The reviewers have made a range of suggestions to complete and improve several figures and tables

DISCUSSION:
* The discussion cannot be a mere repetition of results ; please reinforce the discussion part, in the light of recent publications

More generally, I am ready to consider a carefully revised version of your ms. if you can seriously take into account the points raised by the reviewers. Please then send the revised version along with a point by point response to reviewers' concerns.

Best regards

Xavier

·

Basic reporting

The language is very clear. The authors have used a language editing service, so there is nothing to be said about it. I have marked some minor errors in the attached pdf file.

The structure of the manuscript is ok all background information is there. I believe that the author should add more details on their data analysis in the materials and methods section as it is I can't fully access if the methods used are 100% correct.

The figures and tables are clear and data is available for reanalysis.

Experimental design

The experimental design is ok. The only issue that I find is that the authors have chosen to identify microbial groups that change between plant species using ANOVA. Why not use methods that were designed to do this such as STAMP, metagenomeSeq, Lefse. Also there are details that can change the final result. For instance, was the OTU table normalized? How? Was the p-values of ANOVA corrected with FDR, Boferoni, etc...?

Validity of the findings

The results are not repetition of known facts it is a brand new view of the rhizosphere of these halophites. The experimental design is alright I just think the authors should explain their choice of methods or change it (as stated above.

Additional comments

My general comments are found in the pdf attached.

Reviewer 2 ·

Basic reporting

- The manuscript is poorly written, difficult to follow/understand and contains a certain number of impressions.
- Literature is not always well referenced, in particular in the introduction and discussion many recent paper regarding i) association among bacterial and halophyte and ii) ecological/environmental driver in bacterial assembling/recruitment are not reported.
- The manuscript should be re-written with much more precision and style improvement in order to improve general clarity.
- Figures are not always relevant (see specific comment below in Figure section), moreover they are not well labelled/described.

Experimental design

- Research question and experimental design pf the work is not well defined. Authors not state/identify a specific knowledge gap to be addressed.
- The methodology is not detailed, no enough information are available to replicate the work.
- Only three replicates are used to analyze variability of bacterial community associated to different plant species, additional analysis are necessary to confirm that this number is enough to describe the variability associated to each species.

Validity of the findings

- Conclusions depict only how rhizosphere select different bacterial community then bulk soil. No correlation to chemical analysis/salt concentration/host or link to original research question is reported. Not always the correct statistical analysis has been used.
Role of plant species (or of other factors) was not analyses, described, discussed.

Additional comments

The manuscript proposed by Li et al is describing the bacterial associated to halophytic plants in the salty soil of the Ebinur Lake Nature Reserve (Gurbantungut Desert, China). Rhizospheric soils of five different halophytic plants and their respective bulk soil have been collected in triplicates. To study the bacterial communities 16S rRNA-based amplicon sequencing has been applied to a total of 30 samples. The results obtained showed that the rhizosphere and bulk soils selected specific bacterial communities, with different relative abundance of several taxonomic groups.

Below are listed some general concerns which merit attention:

Abstract:
Not clear background and experimental design was reported. Results/Discussion reported in this section should be underline the main findings, underlining also the importance of the work done and of the results obtained in the frame of plant-bacterial interaction in saline soils.

Introduction:
- Organization of this section is not clear. Concept have been reported and repeated in several point. Not updated bibliography has been provides. Many papers have been recently published on this topic reporting how host and soil parameters (among others salt concentration, tide etc.) can influence assembly and structure of bacterial communities associated to halophytic plants (among others in 2016/2017: doi: 10.1016/j.micres.2016.05.012; 10.1007/s00284-016-1096-7; doi.org/10.3389/fmicb.2016.01286; doi.org/10.3389/fmicb.2017.02288).
- In this section, taxonomic name should be reported in italic.
- Add reference or at least some additional description of the group of plants available in the saline system, they are very different ranging from tree to shrub.
Methods:
- Detail regarding sampling in the field (root depth, soil depth, etc.), preparation of sample (separation of rhizosphere from root) are missed.
- Details regarding the pipeline used for the analysis of the sequence and the obtained OTUS were not well described.
- Line 151, modify ‘mother’ with ‘mothur’.
- Principal components analysis (PCA) cannot be used to analyse matrix of similarity/dissimilarity, as the one obtained by the OTUs table. Please remove this and maintain only the NMDS analysis.
- Please add stress value to the NMDS analysis.
- Line 149. Richness, rarefaction, NMDS are not statistical analysis, please modify title of the paragraph
- Principal components analysis (PCA) (or other adapt to analyze ordinal variable) could be applied to analyse chemical soil analysis matrix (including all the parameters measured) in order to understand how soil change and could influence bacterial composition. It could be associated with an analysis of the factors shaping the bacterial assemblage in order to define which factors drive the bacterial communities (soil parameters vs host). Moreover mantel test could be also useful to depict possible correlation among beta-diversity and soil properties.

Results:
- A sampling scheme (map) in which plant sampled were associated to different environmental parameters (i.e salt concentration, C, N etc.) could be useful to describe/explain the experimental design used.
- Chemical analysis have been conducted on the bulk soil, I not think it was possible do the same for the rhizospheric soil. So please modify the caption of table 1 and clarify this point.
- Line 174 Move NCBI project number in material and method.
- Figure 1A. Please move it in the supplementary material. Calculation of ‘Good’s coverage’ values can be more clear to understand coverage level of sequencing (they can be added in text of methods since are a pre-requisite to perform the other analysis).
- Line 179. It is not clear if chimera and chloroplast sequences have been removed before/after rarefaction analysis. Since these can significantly influence the analysis. I suggest to remove them before to perform analysis of richness, coverage etc. Please specify this point and eventually repeat the analysis.
- Please provide statistical analysis also to evaluate the effect of the factor ‘plant species’. Actually only statistical analysis supporting difference among bulk soil and rhizosphere was provided. Main test and multiple comparison tests should be added to evaluate diversity among plant species, infer is from NMDS graph is not correct (line 203-205).
- Line 237. I suggest to the authors to define it as ‘community composition’. Community structure has been previously reported.
- Line 255. Which are these 50 OTUs? Please define and clarify this last part.
- Line 260. Ven Diagram description could be moved before, in order to better clarify similarity/different depicted by statistical analysis. Please add a ven diagram also for bulk bacterial communities, to underline the different selective process acted by the two categories of samples.

Figure:
- No detailed caption are available, no letter to distinguish panel are present.
- Multi-panel figure to describe beta-diversity/bacterial communities structure could be better for reader understanding
- Figure 3. It is difficult read name of taxa and understand what star (*) mean. In the bottom panel, some names are cut. An additional bar chart with relative abundance (%) of taxa among rhizosphere and bulk soil of the five species could be added as first panel of figure. This will help to better understand the communities composition described in the results.
- Figure 4. Why taxa distribution mong the 5 plant species has been reported only for rhizosphere samples? This visualization is not really clear and easily understandable (see comment before).
- Figure 5. Rank of OTUs list for their taxonomy (i.e phylum, class etc.) could be useful to detect pattern of distribution among the 5 rhizosphere groups. As before I suggest t0 do the same for bulk soil groups, since as stated by the authors the work aim to compare rhizosphere and bulk soil.
- Figure 6. Express the N. of OTUs as percentage of OTUs (and eventually add their relative abundance) could be a more clear way to describe repartition of bacteria among the 5 plant species. Please, do the same for bulk samples.

Table:
- Additional details in table caption are necessary.
- Table 1. Please specify that the reported values are referred to bulk soil (see comment before) where plants growth.
- Table 1 and 2, please add detail regarding statistical analysis used and what letter indicate.
- Supplementary Table 1. Please add all the phylum/class etc. detected, ad indicated with * the significant one. As the author did, multiple-comparison test has been reported for significant different taxa. Multiple-comparison test should be added also for bulk soil samples.
- Supplementary Table 2. Caption, methods and results obtained are not clearly explained in the manuscript. Additional detail and discussion of these data are necessary.

Discussion/Conclusion:
No really discussion and conclusion are present. Not really trend or factor correlations have been reported to explain the results obtained. Repetition of results with lack of discussion in the frame of recent publication (see general comments) is the limitation of this section.

Reviewer 3 ·

Basic reporting

The study by Li and collaborators presents results of a study of the bacterial communities in the rhizosphere associated with 5 halophyte plant species and bulk soil taken at a site with high soil salinity in northwest China. This descriptive study details the communities found in these rhizosphere soils and determine how these communities are either similar or different according to plant species and how they compare to bulk soil. The authors study the results at different taxonomic levels and look at which genera are enriched in either soil type and how they are influenced by the plant species. The work is well written although some errors can be found (see below for some examples). Sufficient background on the topic is provided. The figures are adequate although perhaps the rarefaction curves in Figure 1 could be moved to supplementary material.

Experimental design

The experimental design is adequate for the study proposed by the authors. Salinization of soil is becoming an increasingly important problem for which solutions are needed. PGP rhizobacteria may offer a solution to aid plant growth under these adverse conditions therefore studies like the present with halophyte plant species may help to identify the most interesting bacterial species. With this in mind it would have improved the impact of the study if such plant growth promoting species would have been isolated in the study. Nevertheless, one of the points that need to be improved in the work is a better description of the techniques used. It is not clear from the study how bulk soil was collected. Was it associated with the plants sampled by the author’s or taken close to the sampled plant? If it were associated with the plant roots how did they distinguish between bulk soil and rhizosphere soil? Another problem is that the authors do not give any information about how soil chemical analysis (line 132) was performed.

Validity of the findings

The conclusions reached generally are supported by the results presented except for the role associated with effect of salinity (Abstract line 44; conclusions line 413). No data is given which supports that salinity is an important driver in saline soil. In fact it is in contrast to data discussed on lines 277-279 where soil salinity does neither correlate nor appears to cause differences in bulk soil communities. If the authors wish to determine the effect of salinity additional samples in non-saline soil with the same plants would have to have been studied as well. Alternatively, since it is clear that the EC differences do not appear to affect bulk soil communities a closer look could have been taken only at rhizosphere communities by perhaps performing CCA incorporating EC data. Nevertheless it is surprising that the authors did not present the Pearson correlations in the Results section and only mention it in the discussion as supplementary data. The data mentioned in the text as significantly different (lines 186-195) do not agree with the significance given in Table 2. This needs to be made clearer. Another point that needs to be clarified is the results given in the Venn diagram in figure 6. What do the OTU numbers mean? Were these OTUs shared by all three replicate for each plant species? Probably not because the average number of OTUs for the five halophytes was 1692 (line 186) while the total number of OTUs for each halophyte in the Venn diagram is in the order of 4000 OTUs. This point is important because an OTU found in only one replicate sample is more likely to be shared with that from one rather than by all three from another plant species thereby inflating the number of shared OTUs (line 40; line 261, line 393) which could lead to confusing conclusions (lines 406-409).

Additional comments

1) Fig 1 no figure legend is given to indicate which curves belong to with samples.
2) Examples of errors and sentences which need to be improved:
-Line 92 change was to is
-Line 158 physicochemical
-Line 160 Graphpad
-Line 164 show units for electrical conductance
-Line 173 verb tense
-Line 186, 188 The number of OTUs
-Line 311 separate genusin
-Line 324 change ‘retention’ for reduction
-Line 344 change verb tense
-Line 347 better may be than is
-Line 349 improve ‘matters’
-Line 355 sentence not clear
-Line 406 ANOSIM

---

## Round 0.2 · Major Revisions

You have already addressed many issues that had to be addressed, and have thus improved substantially the ms., which is acknowledged by the reviewers. However, there are 2 remaining, major issues still pending:

* Although the manuscript has apparently been checked by a language editing service and possibly a native speaker (?), the English still has to be significantly improved. An emblematic example is the title where you have used 'co-occurred' instead of 'co-occurring'.
==> I thus ask you to ensure that a proper reviewing of the English of the whole text is made, or this would restrict PeerJ's ability to publish the paper.
==> Please also take into account the remarks and suggestions by the reviewer #3.

* Reviewer #3 also raised a few issues linked to the discussion & conclusions. Please carefully revise these sections accordingly.

Please amend the manuscript. taking into account these last, two important issues, I would be happy to receive the revised version which I hope would be then likely acceptable for publication. Please understand that the objectives of the reviewers and myself are to make sure that your paper presents the results of your work in the best possible way.

best regards
Xavier LE ROUX

·

Basic reporting

The manuscript is well written, clear and fluid. Nothing to mention here.

Experimental design

The authors corrected the issues that I pointed in the previous version.

Validity of the findings

Since the potential mistakes were corrected I don’t see anything to mention here.

Reviewer 3 ·

Basic reporting

The resubmitted version of the manuscript by Li and collaborators generally has succeeded in addressing the reviewer’s comments and is much improved with regard to the previous version. However, at this point the major concern is the quality of the English. Although the manuscript has apparently been checked by a language editing service and possibly a native speaker, with the new writing many new errors have been introduced which cause more confusion instead of helping to clarify notions. The use of ‘co-occurred’ in the title, abstract, and throughout the text as a verb, adjective or without determiner (which or that) is confusing and unnecessary. This needs to be changed, improved or eliminated throughout. Poor or unclear sentences can be found in the Introduction (lines 63-65, lines 68-72, lines 101-103, lines 104-107, lines 116-118), Results (lines 240-241) and Discussion (lines 339-340, lines 444-446), and Conclusions (lines 487-489, lines 489-491). The Discussion section suggested in the author guidelines of PeerJ for the Abstract is missing. In the Abstract and Conclusions it is not mentioned whether ‘much diverse’ (lines 35 and 472) was more or less.

Experimental design

The experimental design is now much clearer and complete.

Validity of the findings

The conclusions reached generally are supported by the results presented. However there are a few minor questions which arise. Now the authors indicate OTUs shared by all three replicates in their Venn diagrams and discuss them briefly (lines 426-432). The conclusion on lines 430-432 is not clear. Also it would be interesting to know more about the high abundance shared OTUs. Which genera do they belong to? Are some of these OTUs included in the taxa found by LefSe (fig 5)? These could be interesting PGPR candidates as successful colonizers of halophyte plants in saline-arid land. On the other hand, although the study is centered on the rhizosphere of halophytes, it would have been interesting to know more of the bulk soil communities with respect to the shared OTUs, LefSe analyses, and the effect of soil properties on these soil communities (CCA). This could have provided some interesting additional information about arid saline soil bacterial communities.

---

## Round 0.3 · accepted · Accept

Dear authors,

The resubmitted version of your manuscript has succeeded in addressing the reviewer’s major and minor comments.

However, please note that some very minor typos remain such as extra spaces, commas or misplaced or lacking periods, non-italicized ‘H’ in ‘Halostachys’, as well as the change of ‘leguminous Rhizobium species’ (Lines 403-404) to ‘the Rhizobium-legume interaction’. Please check this during type setting.

I hope that you have appreciated the review/editorial work done on the manuscript for improving its quality.

best regards
Xavier

# Reviewer 3 ·

Basic reporting

The resubmitted version of the manuscript by Li and collaborators generally has succeeded in addressing the reviewer’s major and minor comments and is much improved with regard to the previous versions. The improved English allows for more fluid reading of the manuscript and a better understanding of the results and conclusions. Some very minor typos remain such as extra spaces, commas or misplaced or lacking periods, non-italicized ‘H’ in ‘Halostachys’ but these as well as the change of ‘leguminous Rhizobium species’ (Lines 403-404) to ‘the Rhizobium-legume interaction’ will surely be changed during type setting.

Experimental design

The experimental design is clear and complete.

Validity of the findings

The conclusions reached are supported by the results presented.

Additional comments

No comment